# RBD-Based ELISA and Luminex Predict Anti-SARS-CoV-2 Surrogate-Neutralizing Activity in Two Longitudinal Cohorts of German and Spanish Health Care Workers

Ruth Aguilar,[a] Xue Li,[b] Claudia S. Crowell,[b] Teresa Burrell,[b] Marta Vidal,[a] Rocio Rubio,[a] Alfons Jiménez,[a,c] Pablo Hernández-Luis,[d,e] Dieter Hofmann,[f,g] Hrvoje Mijočević,[f] Samuel Jeske,[f] Catharina Christa,[f] Elvira D'Ippolito,[b] Paul Lingor,[h] Percy A. Knolle,[g,i] Hedwig Roggendorf,[i] Alina Priller,[i] Sarah Yazici,[i] Carlo Carolis,[j] Alfredo Mayor,[a] Patrik Schreiner,[k] Holger Poppert,[l] Henriette Beyer,[l] Sophia E. Schambeck,[b,l] Luis Izquierdo,[a,m] Marta Tortajada,[n] Ana Angulo,[d,e] Erwin Soutschek,[k] Pablo Engel,[d,e] Alberto Garcia-Basteiro,[a,m,o] Dirk H. Busch,[b,g] Gemma Moncunill,[a,m] Ulrike Protzer,[f,g] Carlota Dobaño,[a,m] Markus Gerhard[b,g]

[a]ISGlobal, Hospital Clínic, Universitat de Barcelona, Barcelona, Catalonia, Spain
[b]Institute of Medical Microbiology, Immunology, and Hygiene, School of Medicine, Technical University of Munich (TUM), Munich, Germany
[c]Centro de Investigación Biomédica en Red de Epidemiología y Salud Pública (CIBERESP), Barcelona, Spain
[d]Immunology Unit, Department of Biomedical Sciences, Faculty of Medicine and Health Sciences, Universitat de Barcelona, Barcelona, Spain
[e]Institut d'Investigacions Biomèdiques August Pi i Sunyer, Barcelona, Spain
[f]Institute of Virology, School of Medicine, Technical University of Munich, Munich, Germany
[g]German Center for Infection Research (DZIF), Munich, Germany
[h]Klinikum rechts der Isar, Department of Neurology, School of Medicine, Technical University of Munich, Munich, Germany
[i]Klinikum rechts der Isar, Institute of Molecular Immunology and Experimental Oncology, School of Medicine, Technical University of Munich, Munich, Germany
[j]Biomolecular Screening and Protein Technologies Unit, Centre for Genomic Regulation (CRG), The Barcelona Institute of Science and Technology, Barcelona, Spain
[k]Mikrogen GmbH, Munich, Germany
[l]Helios Klinikum München West, Munich, Germany
[m]Centro de Investigación Biomédica en Red de Enfermedades Infecciosas (CIBERINFEC), Barcelona, Spain
[n]Occupational Health Department, Hospital Clínic, Universitat de Barcelona, Barcelona, Spain
[o]Department of Preventive Medicine and Epidemiology, Hospital Clinic, Universitat de Barcelona, Barcelona, Spain

Carlota Dobaño and Markus Gerhard contributed equally to this work.

**ABSTRACT** The ability of antibodies to neutralize severe acute respiratory syndrome coronavirus 2 (SARS-CoV-2) is an important correlate of protection. For routine evaluation of protection, however, a simple and cost-efficient anti-SARS-CoV-2 serological assay predictive of serum neutralizing activity is needed. We analyzed clinical epidemiological data and blood samples from two cohorts of health care workers in Barcelona and Munich to compare several immunological readouts for evaluating antibody levels that could be surrogates of neutralizing activity. We measured IgG levels against SARS-CoV-2 spike protein (S), its S2 subunit, the S1 receptor binding domain (RBD), and the full length and C terminus of nucleocapsid (N) protein by Luminex, and against RBD by enzyme-linked immunosorbent assay (ELISA), and assessed those as predictors of plasma surrogate-neutralizing activity measured by a flow cytometry assay. In addition, we determined the clinical and demographic factors affecting plasma surrogate-neutralizing capacity. Both cohorts showed a high positive correlation between IgG levels to S antigen, especially to RBD, and the levels of plasma surrogate-neutralizing activity, suggesting RBD IgG as a good correlate of plasma neutralizing activity. Symptomatic infection, with symptoms such as loss of taste, dyspnea, rigors, fever and fatigue, was positively associated with anti-RBD IgG positivity by ELISA and Luminex, and with plasma surrogate-neutralizing activity. Our serological assays allow for the prediction of serum neutralization activity without the cost, hazards, time, and expertise needed for surrogate or conventional neutralization assays. Once a cutoff is established, these relatively simple high-throughput

Address correspondence to Carlota Dobaño, carlota.dobano@isglobal.org, or Markus Gerhard, markus.gerhard@tum.de.

The authors declare no conflict of interest.

antibody assays will provide a fast and cost-effective method of assessing levels of protection from SARS-CoV-2 infection.

**IMPORTANCE** Neutralizing antibody titers are the best correlate of protection against SARS-CoV-2. However, current tests to measure plasma or serum neutralizing activity do not allow high-throughput screening at the population level. Serological tests could be an alternative if they are proved to be good predictors of plasma neutralizing activity. In this study, we analyzed the SARS-CoV-2 serological profiles of two cohorts of health care workers by applying Luminex and ELISA in-house serological assays. Correlations of both serological tests were assessed between them and with a flow cytometry assay to determine plasma surrogate-neutralizing activity. Both assays showed a high positive correlation between IgG levels to S antigens, especially RBD, and the levels of plasma surrogate-neutralizing activity. This result suggests IgG to RBD as a good correlate of plasma surrogate-neutralizing activity and indicates that serology of IgG to RBD could be used to assess levels of protection from SARS-CoV-2 infection.

**KEYWORDS** SARS-CoV-2, neutralization, antibodies, immunoglobulin G, spike protein, receptor binding domain, ELISA, Luminex, symptoms

Since the start of the COVID-19 pandemic, the surveillance of antibody-seropositive individuals has been an important measure of the spread of infection and the prevalence of exposure in the population. Health care workers (HCW) face a higher risk of exposure to severe acute respiratory syndrome coronavirus 2 (SARS-CoV-2) due to their direct patient contact (1, 2), and several studies have monitored the spread of the infection and the dynamics of immune responses in longitudinal cohorts of HCW (3–8). There is, however, no current standardization of assays designed to measure antibodies to SARS-CoV-2. Studies have thus far used different approaches, including antibody binding and functional assays, with varying sensitivities and specificities, making it difficult to compare seroprevalence rates across study populations. As such, the World Health Organization (WHO) has called for the harmonization and standardization of SARS-CoV-2 serological assays (9). It is imperative that candidate assays not only be easily deployable in a variety of settings but also be confirmed by reliable surrogates of immune protection to allow for high-throughput screening of plasma or serum samples to assess the level of immunity following infection and vaccination.

Data suggest that neutralizing antibody (nAbs) titers are the best correlate of protection against SARS-CoV-2 (10, 11) and that the receptor binding domain (RBD) of the spike (S) protein is the major target for nAbs (12–14). S1 and S2 antigens are contained in first-generation vaccines (15). Insight into the persistence of SARS-CoV-2 nAbs in plasma or serum of convalescent individuals recovered from COVID-19 or vaccinated individuals is critical for understanding population seroprevalence and protection (16) and for assessing the need for additional vaccine doses. The measurement of nAbs is also relevant in the context of passive immunization in the screening of candidate plasma from convalescent donors (17, 18).

Currently, there are three types of tests that can be used for measuring nAbs. The conventional infection neutralization test using real virus is the gold standard; however, it requires handling live SARS-CoV-2 virus in a specialized biosafety level 3 containment facility and takes 2 to 4 days to complete (16, 19). Pseudovirus-based neutralization tests have been established that use viral vectors expressing a marker gene and are pseudotyped with SARS-CoV-2 S protein. They can be performed more easily but still require a biosafety level 2 laboratory (20). Finally, tests that measure the capacity of plasma or serum antibodies to inhibit the binding of purified RBD to its host cell receptor angiotensin converting enzyme-2 (ACE2) can be performed as surrogate neutralization assays in any laboratory in an enzyme immunoassay plate (21–23), using chemiluminescence immunoassays (24) or by cytometry (25). However, high-throughput screening of plasma or serum samples is

**TABLE 1** Characteristics of the study participants

| Characteristic | Munich (*n* = 218) | Barcelona (*n* = 578) |
|---|---|---|
| Age (yr, mean ± SD) | 40.38 ± 12.44 | 42.10 ± 11.57 |
| Sex (no. [%]) | | |
| Male | 87 (39.91) | 161 (27.85) |
| Female | 131 (60.09) | 417 (72.15) |
| Visit times (no. [%])[a] | | |
| 1 | 95 (43.58) | 12 (2.08) |
| 2 | 66 (30.27) | 55 (9.53) |
| 3 | 8 (3.67) | 446 (77.16) |
| 4 | 10 (4.59) | 65 (11.25) |
| 5 | 39 (17.89) | |
| Direct contact with COVID-19 patients (no. [%]) | | |
| Yes | 130 (59.63) | 455 (78.72) |
| No | 85 (38.99) | 123 (21.28) |
| Missing | 3 (1.38) | |
| COVID-19 typical symptoms (no. [%]) | | |
| Yes | 126 (57.80) | 210 (36.33) |
| No | 83 (38.07) | 368 (63.67) |
| Missing | 9 (4.13) | |
| rRT-PCR test for diagnosis (no. [%]) | | |
| Positive | 73 (33.49) | 39 (6.75) |
| Negative | 35 (16.05) | 248 (42.9) |
| Missing | 110 (50.46) | 291 (50.35) |

[a]Visit times refer to the number of samples from different time points we have for each participant. Thus, in the Barcelona cohort, most of the participants had three samples, while in the Munich cohort, most of the participants had 1 or 2 samples.

not easily achieved with any of these methods, and less expensive, simple, rapid, and reproducible surrogate assays are needed.

We hypothesized that, as RBD is the major target for nAbs, antibody levels to this fragment of S protein would be a good alternative to predict serum neutralization function. In this study, we combined clinical and epidemiological data from two separate cohorts of HCW and exchanged plasma samples with a range of anti-SARS-CoV-2 reactivity between two labs to test the comparability of two immunological readouts for evaluating antibody levels that could predict neutralizing activity and protection. Specifically, we assessed IgG levels against S, S2, RBD, and nucleocapsid (N) full length and C terminus measured by a Luminex assay developed at ISGlobal in Spain, and IgG against RBD measured by an enzyme-linked immunosorbent assay (ELISA) developed at Technical University of Munich (TUM) and Mikrogen GmbH in Germany. We compared these assays to surrogate plasma neutralizing activity measured by a flow cytometry cell-based neutralization assay developed at IDIBAPS/UB in Spain. In addition, we assessed the association between clinical and demographical factors and convalescent plasma or serum surrogate-neutralizing capacity.

## RESULTS

**Clinical, demographic, and serological characteristics of study participants.** A higher proportion of the study participants were female, 60% (131/218) in the Munich cohort and 72.1% (417/578) in the Barcelona cohort (Table 1). Real-time reverse transcription-PCR (rRT-PCR) data were available for approximately 50% of both cohorts; 67.6% (73/108) were positive in the Munich cohort and 13.6% (39/287) in the Barcelona cohort. From the participants with clinical data available, 60.3% (126/209) presented symptoms compatible with COVID-19 in the Munich cohort and 36.3% (210/578) in the Barcelona cohort (Table 1). The proportion of SARS-CoV-2 seropositive

**TABLE 2** Serology data of the study participants at the different visits

| Visit according to cohort (mo) | LUMINEX serology (IgG, IgM, IgA against S, S2, RBD, N-FL, and N-Cterm) | | | |
|---|---|---|---|---|
| | No. positive (%) | No. indeterminate (%) | No. negative (%) | No. missing (%) |
| Barcelona | | | | |
| M0 | 80 (13.8) | 51 (8.8) | 447 (77.3) | 0 (0) |
| M1 | 83 (14.7) | 59 (10.4) | 422 (74.7) | 1 (0.2) |
| M3[a] | 44 (62.9) | 4 (5.7) | 15 (21.4) | 7 (10.0) |
| M6 | 85 (16.8) | 35 (6.9) | 387 (76.3) | 0 (0) |
| | | | | |
| Munich | | | | |
| M0 | 133 (61) | 14 6.4) | 53 (24.3) | 18 (8.3) |
| M1 | 45 (77.6) | 3 (5.2) | 9 (15.5) | 1 (1.7) |
| M2 | 52 (75.4) | 6 (8.7) | 9 (13) | 2 (2.9) |
| M3 | 77 (82.8) | 4 (4.3) | 12 (12.9) | 0 (0) |
| M6 | 26 (54.2) | 3 (6.3) | 4 (8.3) | 15 (31.2) |

[a]Only participants with any previous evidence of SARS-CoV-2 infection were invited to participate at study M3 (8).
M, month.

participants in each study cohort at each of the follow-up visits was determined by the Luminex assay using the three isotypes (IgG, IgA, and IgM) (Table 2). Prevalence of seropositivity was higher in the Munich cohort than in the Barcelona cohort (Table 2). Levels of several antibodies were higher in the symptomatic participants from the Munich cohort than in the Barcelona cohort, especially IgG and IgA to several antigens at recruitment and 1 and 6 months later (Fig. 1A). An opposite pattern was observed for IgG in the asymptomatic participants, with higher levels in the Barcelona compared to the Munich cohort across all time points (Fig. 1B).

**Correlations of antibody levels with plasma surrogate-neutralizing activity.** We first assessed the correlations of the different antibodies measured by Luminex with plasma surrogate-neutralizing activity, all data together (combined cohort) and stratifying by study cohort (Fig. 2). IgG to S antigens showed the highest correlations (Spearman's correlation, rho = 0.4 to 0.6, $P < 0.001$) with rho = 0.6 for anti-RBD IgG in the combined cohort, rho = 0.58 for anti-RBD IgG in the Munich cohort, and rho = 0.6 for IgG-S and anti-RBD IgG in the Barcelona cohort. IgA to S antigens showed low to moderate correlations with plasma surrogate-neutralizing activity depending on the cohort and antigen (rho = 0.3 to 0.48, $P < 0.05$), with the highest rho of 0.48 for IgA-RBD in the Munich cohort, rho = 0.45 for IgA-S and 0.47 for IgA-S2 in the Barcelona cohort, and rho = 0.45 for IgA-S and 0.46 for IgA-S2 in the combined cohort. IgM did not show any correlation with plasma surrogate-neutralizing activity in the Munich cohort, but IgM to S antigens showed low correlations in the Barcelona cohort (rho = 0.26 to 0.28, $P < 0.0001$).

We also checked the correlations excluding negative samples (see Table S1 in the supplemental material), and results were similar with few exceptions.

Levels of IgG against RBD measured by ELISA showed a strong correlation with anti-RBD IgG levels measured by Luminex (rho = 0.78 for samples from the combined cohort, rho = 0.88 for Barcelona samples, and rho = 0.74 for Munich samples; $P < 0.0001$) (Fig. 3A). Accordingly, anti-RBD IgG levels measured by ELISA also showed a good correlation with plasma surrogate-neutralizing activity (rho = 0.64, $P < 0.0001$ for the combined cohort; rho = 0.67, $P < 0.0001$ for the Barcelona samples; and rho = 0.4, $P = 0.004$ for Munich samples) (Fig. 3B). These correlations were also performed excluding the negative samples showing similar results (see Table S2 in the supplemental material).

**Prediction of plasma surrogate-neutralizing activity through antibody levels.** Linear regression models were constructed to predict plasma surrogate-neutralizing activity through anti-RBD IgG levels measured by ELISA or Luminex. Anti-RBD IgG levels measured by ELISA and Luminex were significantly associated with plasma surrogate-neutralizing activity ($P < 0.001$) (Table 3). Participants' plasma surrogate-neutralizing activity increased 9.16% for each unit of anti-RBD IgG levels measured by ELISA

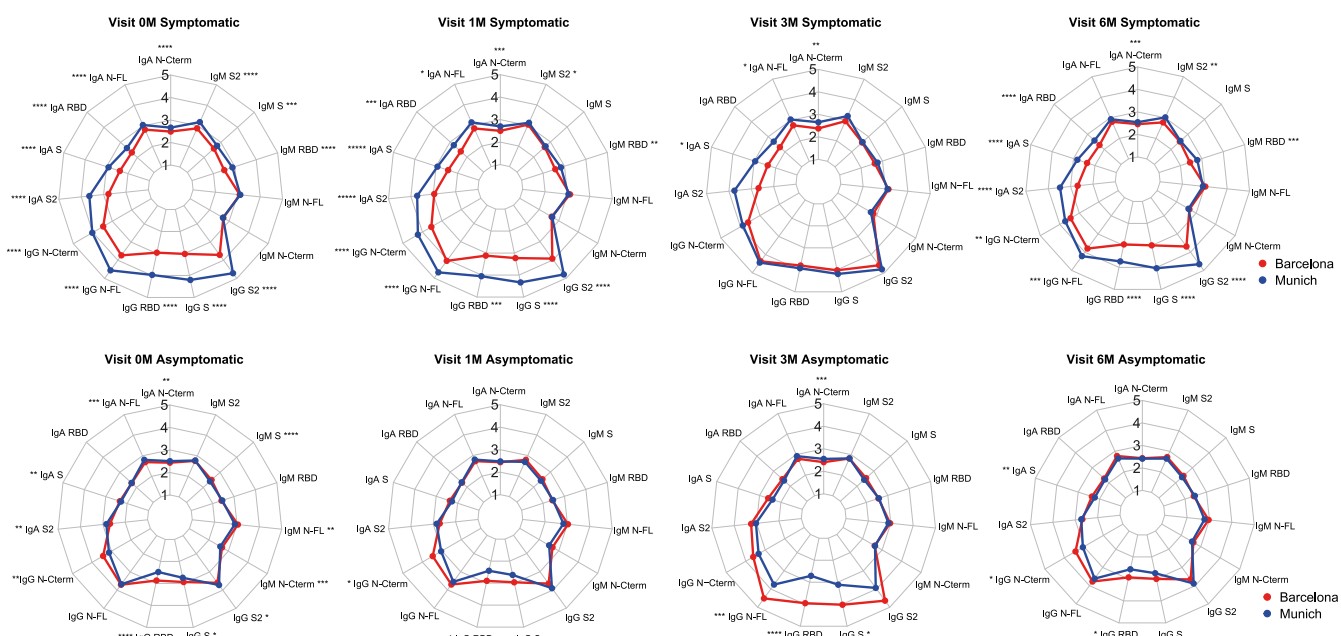

**FIG 1** Radar charts comparing antigen-specific antibody levels (IgG, IgA, and IgM) between Munich and Barcelona cohorts in symptomatic (A) and asymptomatic (B) participants at the different sampling time points. The dots represent the median values of $\log_{10}$ MFI antibody levels. Wilcoxon rank sum test was used to assess statistically significant differences in antibody levels. *, $P < 0.05$; **, $P < 0.01$; ***, $P < 0.001$. Samples sizes in the symptomatic group, 0 months ($n = 336$), 1 month ($n = 123$), 3 months ($n = 79$), and 6 months ($n = 131$). Samples sizes in the asymptomatic group, 0 months ($n = 451$), 1 month ($n = 499$), 3 months ($n = 84$), and 6 months ($n = 424$).

(Table 3), and plasma surrogate-neutralizing activity increased 14.25% for each unit of anti-RBD IgG levels measured by Luminex (Table 3).

IgG levels measured by Luminex against RBD, the N full length (N-FL), the N C-terminal region (N-Cterm), S and S2 proteins were combined to predict plasma surrogate-neutralizing activity. A multiple linear regression model was constructed with anti-RBD, anti-N-FL, anti-N-Cterm, anti-S and anti-S2 antibodies as initial variables and optimized by backward selection. IgG against N-Cterm was excluded from the model because of nonsignificant association with surrogate-neutralizing activity (data not shown). IgG-N-FL and IgG-S2 levels were significantly negatively associated with plasma surrogate-neutralizing activity. Anti-RBD IgG and anti-S IgG levels were significantly positively correlated with plasma surrogate-neutralizing activity (Table 4). The relative importance of each antigen-specific IgG included in the model was 37.17% for anti-RBD IgG, 30.66% for IgG-S, 17.55% for IgG-S2, and 14.62% for IgG-N-FL of the $R^2$, respectively (Fig. 4). The prediction model showed good accuracy with a root mean square error (RSME) of 12.57% through leave-one-out cross-validation (LOOCV).

**Factors associated with anti-RBD IgG levels and plasma surrogate-neutralizing activity.** Multiple linear regression models adjusted for age and sex were used to assess the factors associated with anti-RBD IgG levels measured by ELISA or Luminex and factors associated with plasma surrogate-neutralizing activity. Presence of COVID-19-compatible symptoms, specifically fatigue, loss of smell and taste, dyspnea, headache, rigors, fever, cough, rhinorrhea, sore throat, diarrhea, and the use of medications, specifically acid blockers, were significantly positively associated with anti-RBD IgG levels measured by ELISA and Luminex (Fig. 5A). Loss of taste, dyspnea, rigors, fever, and fatigue were significantly positively associated with surrogate-neutralizing activity (Fig. 5B). Among the subgroup of participants with a positive SARS-CoV-2 rRT-PCR result at the time of recruitment, anti-RBD IgG levels were significantly higher in symptomatic versus asymptomatic participants for the first 3 to 4 months (Fig. 6), but no differences were observed in the plasma surrogate-neutralizing capacity (Fig. 7A), likely due to the small sample size. In this same subgroup of rRT-PCR positives, after stratifying by the symptoms positively associated with surrogate neutralization in the models,

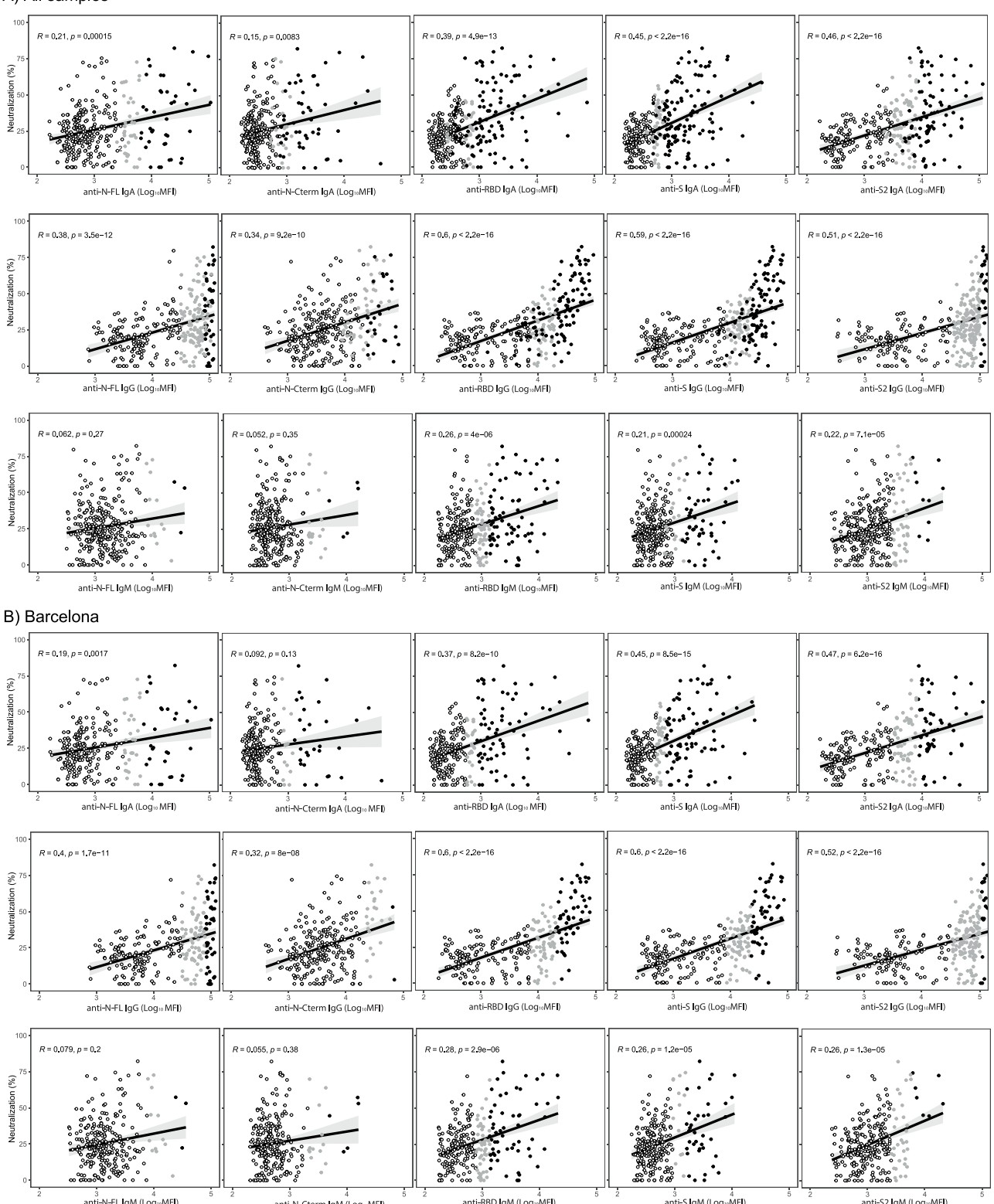

## Luminex serology

● Positive ● Indeterminate ○ Negative

**FIG 2** (Continued)

C) Munich

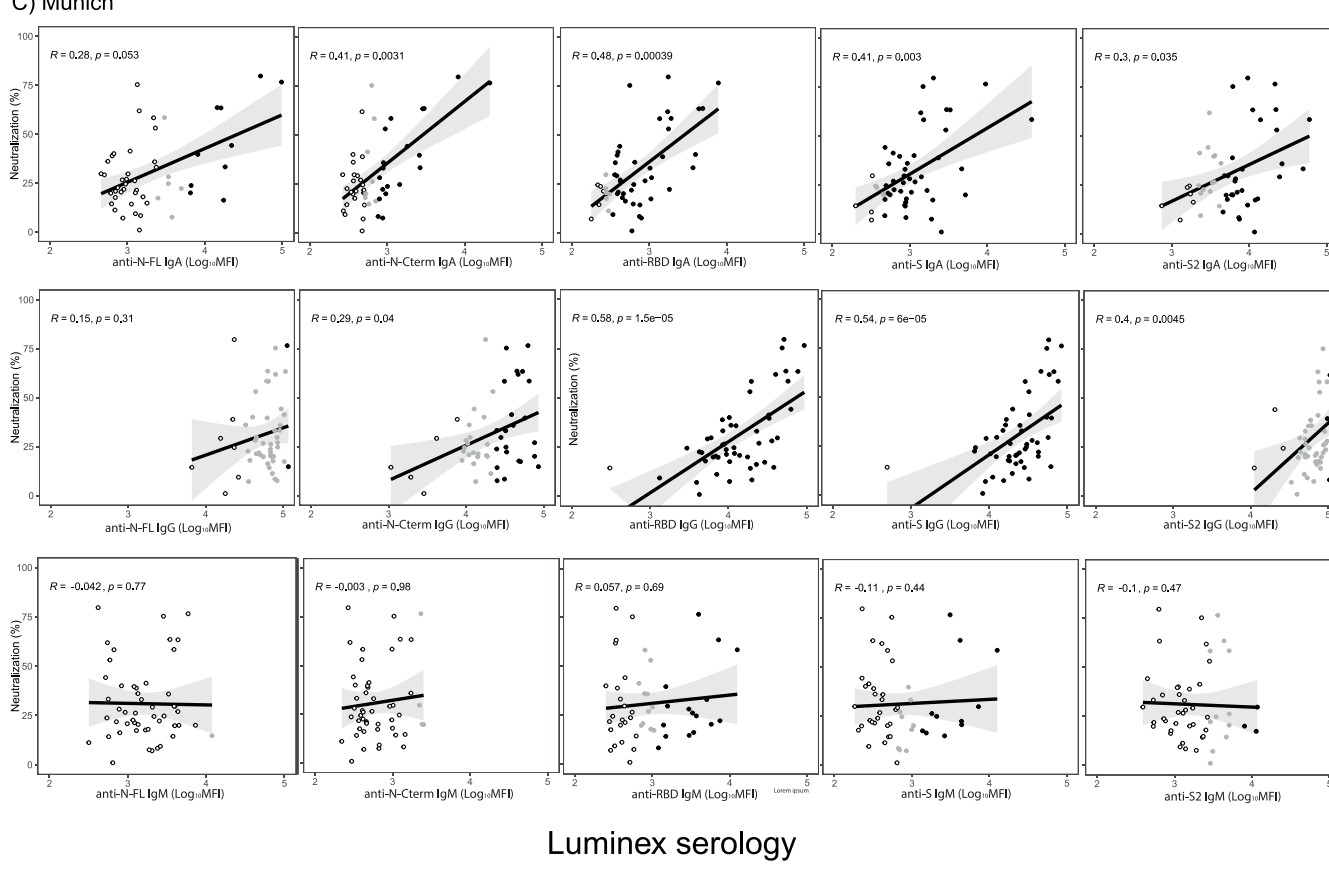

**FIG 2** Correlations between antibody levels measured by Luminex and RBD-ACE2 neutralization capacity. Spearman's rank correlation test between levels ($\log_{10}$ of median fluorescence intensity [MFI]) of IgA, IgG, and IgM against each antigen (nucleocapsid full-length protein [N-FL], its C-terminal region [N-Cterm], the receptor binding domain [RBD], the full S protein, and its subregion S2) and plasma neutralizing capacity (as a percentage of RBD-ACE2 binding inhibition). Data are presented for both sites together ($n = 315$ samples) and stratified by site (Barcelona, $n = 265$; Munich, $n = 50$). Each dot represents an individual measurement. Black lines represent the fitted curve calculated using the linear model method. Shaded areas represent 95% confidence intervals. Black, gray, and white dots correspond to positive, indeterminate, and negative MFIs, respectively. Indeterminate results correspond to MFIs between the positivity threshold (10 to the mean plus 3 standard deviations [SD] of $\log_{10}$-transformed MFI of the prepandemic controls) and an upper limit at 10 to the mean plus 4.5 SD of the $\log_{10}$-transformed MFIs of prepandemic samples.

subjects with dyspnea or rigors showed higher plasma surrogate-neutralizing activity at baseline compared to participants without any of these symptoms, but differences disappeared 1 month later, again likely due to the small sample size, and specially at later time points. No significant differences were observed for loss of taste, fatigue, and fever, except higher surrogate-neutralizing activity levels for participants without fever 1 month after infection (Fig. 7B).

## DISCUSSION

In this study, we compared clinical, demographic, and serological data from two HCW cohorts followed since the first wave of the COVID-19 pandemic in Spring 2020 in Barcelona and Munich. We exchanged 796 plasma samples to apply the respective Luminex and ELISA in-house serological assays and assessed the correlations between each assay and a flow cytometry cell-based assay to determine plasma surrogate-neutralizing activity. Among the study participants, some were SARS-CoV-2 convalescent individuals with mild to moderate disease, and some did not report SARS-CoV-2 infection. Our data showed different serological profiles between cohorts, with higher prevalence and levels of anti-SARS-CoV-2 antibodies in the Munich cohort than in the Barcelona cohort. However, both cohorts showed a high positive correlation between

IgG levels to S antigens, especially RBD, and the levels of plasma surrogate-neutralizing activity, suggesting IgG to RBD as a good correlate of plasma surrogate-neutralizing activity in different study populations.

Differences in the serological profiles between cohorts may be related to differences in (i) the recruitment criteria, (ii) the baseline characteristics of the participants such as the prevalence of comorbidities and/or immunological status, and (iii) the prevalence of moderate to severe cases among the symptomatic cases. Each of these factors may affect antibody levels at recruitment and across follow-up time points. Despite these intercohort differences, the correlation of anti-RBD IgG levels with nAbs was similarly strong in both cohorts.

Previous studies have shown that plasma/serum concentrations of RBD- and S-specific IgG correlate with different neutralization assays (26–31), including several months after infection (32). This is not surprising, as much of the immunodominant response associated with neutralization involves binding and/or blocking the S RBD to inhibit viral entry to host cells (12, 14). Interestingly, IgA binding S antigens also showed low to moderate correlations with plasma surrogate-neutralizing activity, indicating a relevant role of this antibody in the elimination of the virus through RBD (33). Indeed, three consecutive S antigen exposures, either two vaccinations in convalescents or breakthrough infection of twice-vaccinated, resulted in an increasing neutralization capacity per anti-S antibody unit (34). Interestingly, IgG-N-FL and IgG-S2 levels were significantly inversely associated with plasma surrogate-neutralizing activity. The nAb assay used in the study is RBD-based, and N and S2 do not contain RBD, thus higher antibody levels against N and S2 were not expected to increase the percentage of neutralization. However, we do not have an explanation for the negative correlations observed. They could be interfering with the anti-RBD response or indirectly associated to a lower neutralizing activity.

The presence of several COVID-19-compatible symptoms, specifically loss of taste and smell, fatigue, rigors, fever, dyspnea, cough, headache, sore throat, diarrhea and rhinitis were positively associated with anti-RBD IgG levels. Presence of COVID-19-compatible symptoms has been previously associated to higher anti-RBD IgG levels that increase with the number of symptoms (35). Dyspnea (36), fever (37), fatigue (38), and diarrhea (29) have been previously associated with higher anti-RBD IgG levels in COVID-19 convalescent plasma donors. These symptoms can be signs of a systemic inflammatory response, which is likely key for developing a strong anti-SARS-CoV-2 antibody response. Loss of smell and taste have also been associated with higher anti-RBD IgG levels (29); however, the relationship between loss of smell and loss of taste and antibody titers is likely indirect, and the main driving factors remain to be elucidated. It is possible that these symptoms are proportional to the viral load, and thus indirectly associated with the antibody levels.

The use of medications, specifically acid blockers, was associated with higher anti-RBD IgG levels. Many acid blockers are histamine type 2 receptor antagonists with a reported potential therapeutic role for SARS-CoV-2 (39–41). Histamine type 2 receptor antagonists restrict a papain-like protease of SARS-CoV-2 that is essential to the entry of the virus into the host body (42). This antiviral therapeutic role could be indirectly associated with a higher production of anti-RBD IgG levels through a longer exposure of the virus to the immune system outside the host cells.

Loss of taste, dyspnea, rigors, fever, and fatigue were also positively associated with plasma surrogate-neutralizing activity. Previous studies have reported higher neutralizing function in subjects with similar symptoms (31, 43–45). Presence of symptoms is associated with higher levels of antibodies, which are associated with higher neutralizing activity. However, no differences were observed in the subset of participants with a positive rRT-PCR, only for dyspnea and rigors at baseline, likely due to the small sample size, especially at later time points.

One limitation of the study is the difference in the recruitment criteria for the Munich and Barcelona cohorts, making it difficult to explain the reasons for their different serological profiles. A second limitation is the lack of data on neutralizing activity in infection assays and also surrogate-neutralization activity for many of the study

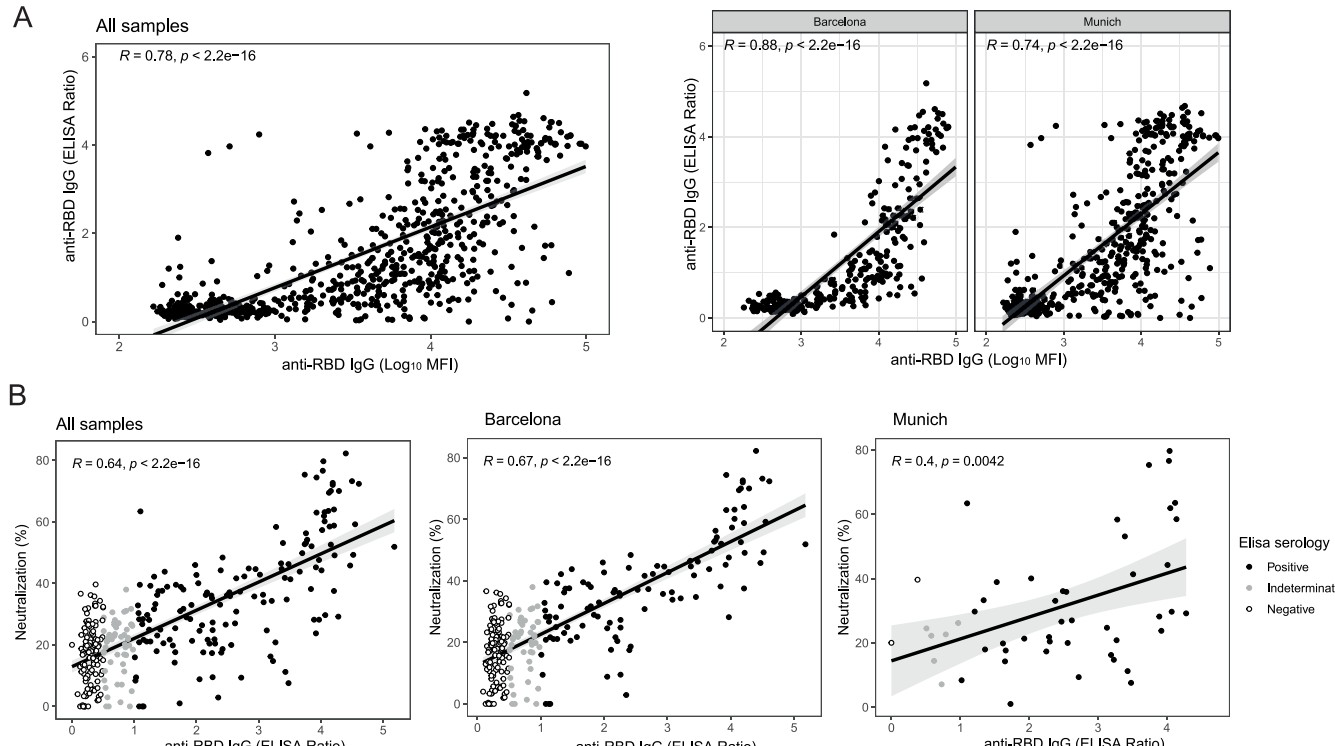

**FIG 3** Correlations of IgG-RBD levels measured by Luminex and ELISA between them and by ELISA with plasma neutralizing activity. (A) Correlation between IgG-RBD levels measured by Luminex and ELISA ($n$ = 710 samples, 266 from Barcelona and 444 from Munich). (B) Correlation between IgG-RBD levels measured by ELISA and plasma neutralization activity ($n$ = 315 samples, 265 from Barcelona and 50 from Munich). Data is presented for both sites together and stratified. Each dot represents an individual measurement. Black lines represent the fitted curve calculated using the linear model method. Shaded areas around the regression lines represent the 95% confidence intervals of the mean. Black, gray and white dots correspond to positive, indeterminate, and negative ELISA measurements, respectively. Indeterminate results in ELISA data correspond to a 0.5 to 1.0 ratio.

samples; however, due to budget constrains, we could only perform this test in a subset of participants. A third limitation is the low sample size in the analyses with the subgroup of participants with rRT-PCR-positive results and neutralization data, which may have caused missing associations. However, low sample size did not affect the analyses with all the study participants with neutralization data; thus, our main conclusion that serology of IgG to RBD is a good surrogate of percent neutralization was not affected.

Although there is no common reference assay for SARS-CoV-2 serology worldwide and correlations between in-house and commercial ELISA- or Luminex-based assays and neutralization activity may slightly differ between laboratories depending on reagents and procedures used, it is remarkable that results are consistent for RBD IgG. To improve comparability among surrogate virus neutralization assays from different origins, it is critical to establish international units per milliliter (IU/mL) for SARS-CoV-2 serum/plasma neutralizing activity. Since the performance of our assays was very robust, we are confident that they can accurately be used as surrogates of neutralizing activity despite the slight differences that there might be with other similar manufactured assays.

**TABLE 3** Linear regression models to assess the associations of IgG-RBD levels measured by ELISA and Luminex with plasma neutralization activity

| | Percent neutralization | | |
|---|---|---|---|
| Predictors | Estimates | 95% CI | P value |
| Intercept[a] | 12.94 | 10.98 to 14.89 | <0.001 |
| Anti-RBD IgG ELISA (ratio) | 9.16 | 8.20 to 10.12 | <0.001 |
| Intercept[a] | −25.74 | −33.56 to −17.93 | <0.001 |
| Anti-RBD IgG Luminex (log$_{10}$MFI) | 14.25 | 12.17 to 16.34 | <0.001 |

[a]Intercept, the value of neutralization when IgG to RBD is equal to zero.

**TABLE 4** Multiple linear regression model to assess the association of IgG levels measured by Luminex against RBD, N-FL, S, and S2 with plasma neutralization activity

| Predictors | Percent neutralization | | |
| --- | --- | --- | --- |
| | Estimates | 95% CI | P value |
| Intercept[a] | 9.75 | −1.29 to 20.78 | 0.083 |
| Anti-RBD IgG (log$_{10}$MFI) | 21.79 | 15.80 to 27.78 | <0.001 |
| Anti-N-FL IgG (log$_{10}$MFI) | −16.12 | −21.50 to −10.75 | <0.001 |
| Anti-S IgG (log$_{10}$MFI) | 16.78 | 9.80 to 23.76 | <0.001 |
| Anti-S2 IgG (log$_{10}$MFI) | −13.14 | −18.80 to −7.48 | <0.001 |

[a]Intercept, the value of neutralization when IgG to RBD is equal to zero.

The correlations that we observe between anti-RBD IgG levels and surrogate-neutralizing activity of sera or plasma from both cohorts, and by Luminex and ELISA, indicate that our assays measure relevant antibodies without the cost, hazards, time, and expertise needed for neutralizing assays. An additional advantage is that RBD from variants of concern (VoC) is relatively easy and rapid to produce. Thus, these antibody assays can be quickly adapted to measure plasma surrogate-neutralizing activity against RBD from newly emerging variants (46). nAbs binding the RBD are considered a serological correlate of protection; thus, once a cutoff is established, these simple antibody assays may allow for assessment of levels of protection to VoC (11, 47). The main limitation of these assays is that they only measure antibodies binding the RBD epitopes. Thus, antibodies binding the N terminus of S, which have also been reported to be neutralizing, cannot be determined. As such, an RBD-focused assay is unable to account for additive action of antibodies targeting different epitopes (48, 49), which is very important to assess with the increasing prevalence of immune escape VoC (50, 51). To compensate for this limitation, the association of specific symptoms with higher levels of neutralizing antibodies could be used in combination with the measurement of RBD IgG levels to develop predictive models to estimate the probability and magnitude of the nAbs response. These models could be useful for better estimating the actual population immunity level in regions without the facilities and technology needed to perform plasma neutralization assays.

## MATERIALS AND METHODS

**Munich cohort.** From April to October 2020, 4,554 HCW of the University Hospital München rechts der Isar of TUM (Germany) and affiliated teaching hospitals were recruited within the EPI-SARS ("Establishment and validation of epitope-specific SARS-CoV-2 blood-based testing methods") and SeCoMRI (52) studies. These studies were extended for further follow-up after BNT162b2 mRNA vaccination (VaCoMRI). Both studies were approved by the ethics committee of the University Hospital rechts der Isar of TUM (ethics vote 182/20S, 216/20S, 476/20S, and 26/21S-SR). Written informed consent was obtained from all study participants at enrollment. Anonymized serum samples were obtained from the biobank of the University Hospital rechts der Isar (ethics protocol 216/20S). Participants with clinical symptoms compatible with COVID-19 illness, or who had a positive SARS-CoV-2 nasopharyngeal rRT-PCR result regardless of the

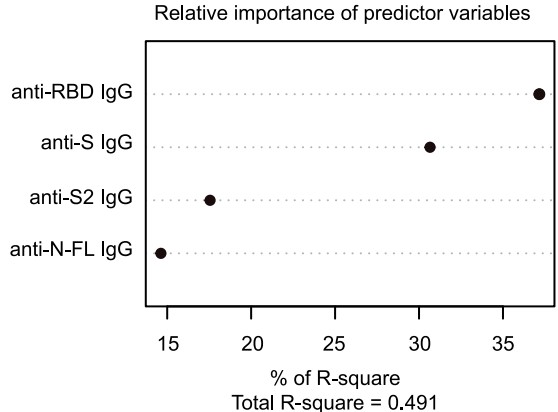

**FIG 4** Relative importance of each predictor variable for the $R^2$ in the multiple linear regression model assessing the association of IgG against RBD, N-FL, S, and S2 with plasma neutralization activity.

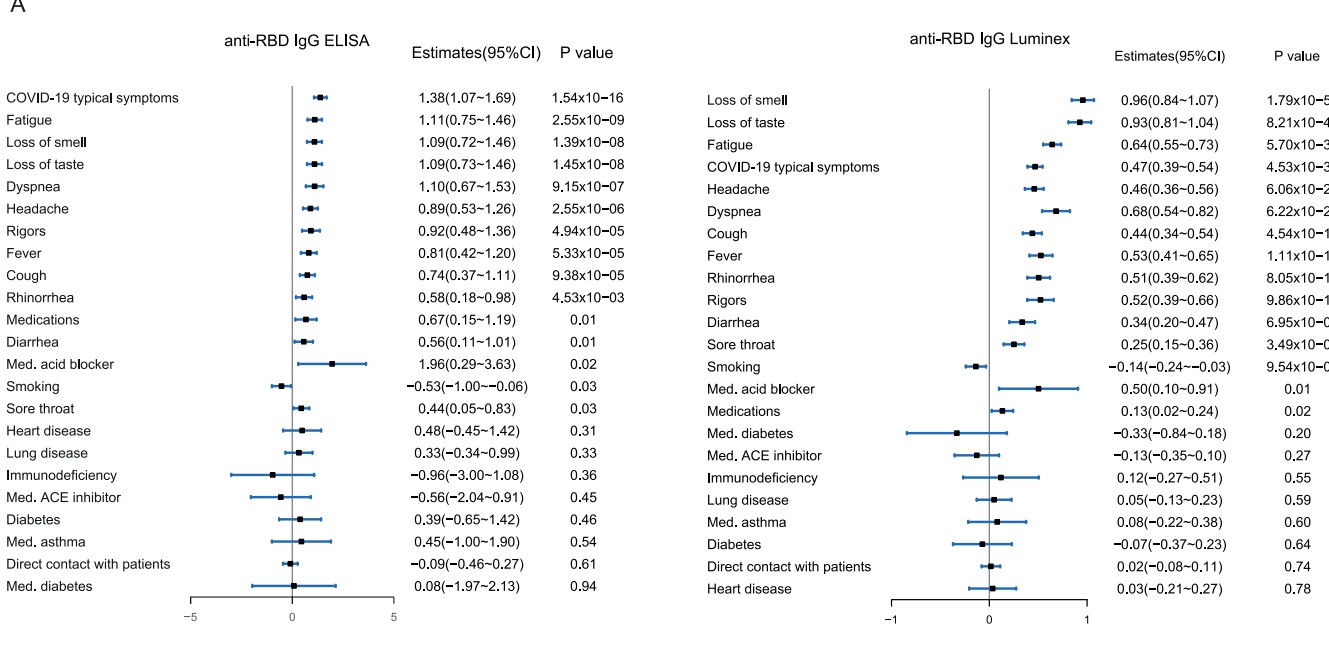

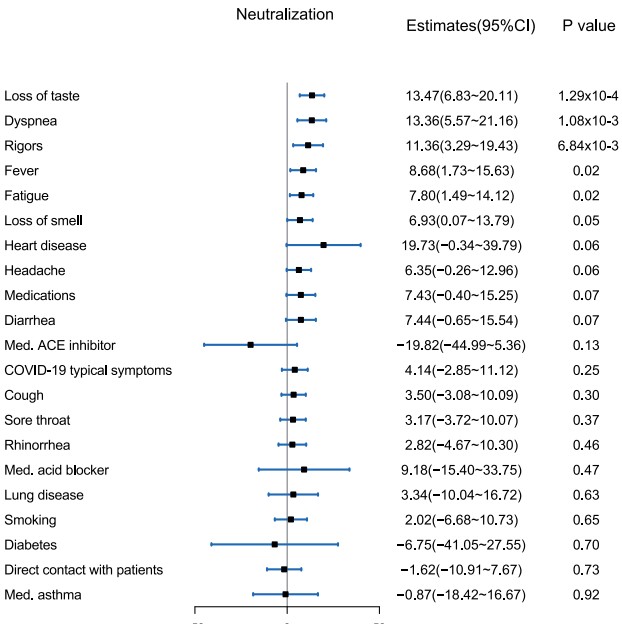

**FIG 5** Forest plots showing the factors associated to IgG-RBD levels measured by ELISA and Luminex (A) and plasma neutralization activity (B). Sample size in panel A is 302 for the anti-RBD IgG ELISA and 778 for the anti-RBD IgG Luminex. Sample size in panel B is 111 for the neutralization data. Multiple linear regression models were used, and all estimates were adjusted for age and sex. The dots represent the point estimates, while the whiskers depict the corresponding 95% confidence intervals.

presence of symptoms, were included in this substudy, with a total of 218 at baseline (study month 0 [M0]). Participants were followed over a period of 6 months with follow-up visits at months 1 (M1), M2, M3, and M6 to assess clinical status and obtain serial blood samples by venipuncture. None of the participants from the Munich cohort had been vaccinated at any of the time points tested.

**Barcelona cohort.** A total of 578 randomly selected HCW were recruited during the first wave of the COVID-19 pandemic in spring 2020 in the Hospital Clínic de Barcelona in Spain within the SEROCOV study. Protocol and informed consent were approved by the local ethics committee of the Hospital Clínic de Barcelona (CEIm) (reference number HCB/2020/0336). Written informed consent was obtained from all study participants at enrollment. Participants were recruited in spring 2020 (study month 0 [M0]) and performed two additional study visits at M1 and M6 (25). Participants with any previous evidence of SARS-CoV-2 infection were also invited to follow up at M3 (8). rRT-PCR tests were performed at

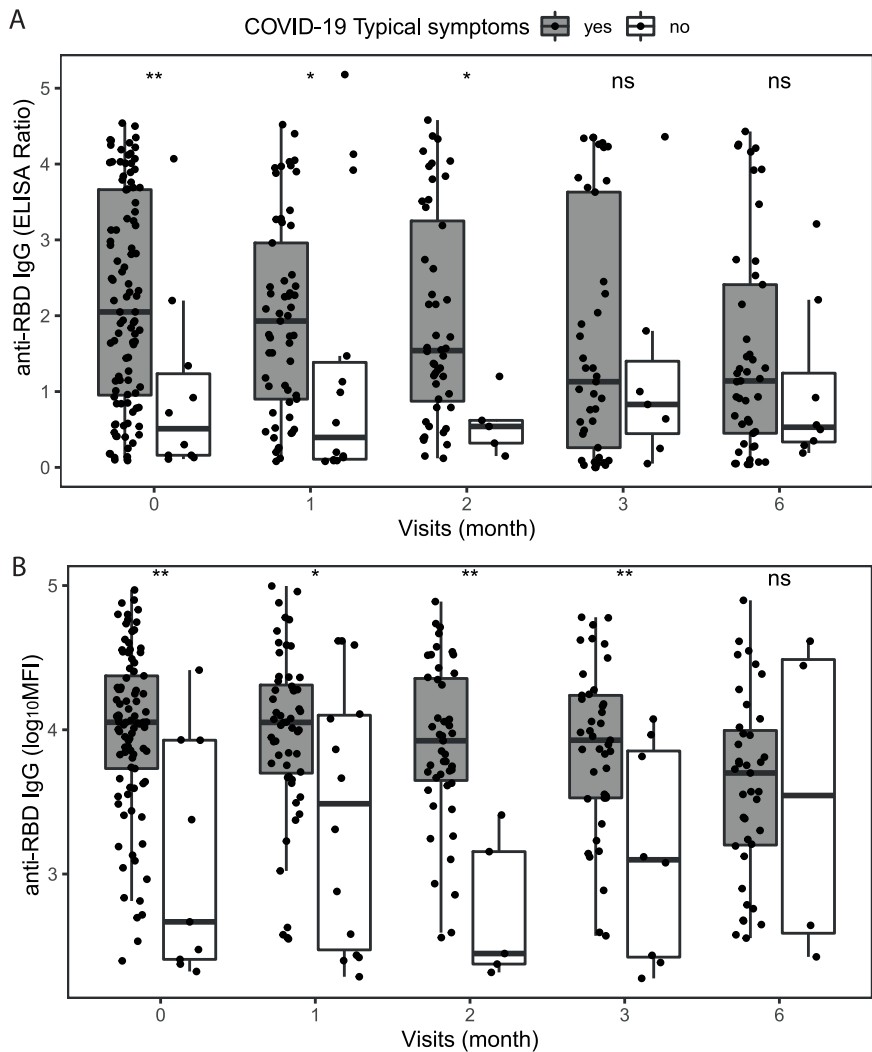

**FIG 6** Comparison of IgG-RBD levels between symptomatic and asymptomatic individuals at different time points after a positive diagnostic by rRT-PCR. (A) IgG-RBD measured by ELISA ($n = 335$ samples). (B) IgG-RBD measured by Luminex ($n = 316$ samples). Plots show IgG-RBD levels measured as ratio for ELISA and $\log_{10}$ MFI for Luminex data. The center line of boxes depicts the median values, the lower and upper hinges correspond to the first and third quartiles, the distance between the first and third quartiles corresponds to the interquartile range (IQR), and the whiskers extend from the hinge to the highest or lowest value within $1.5\times$ IQR of the respective hinge. Wilcoxon rank sum test was used to assess statistically significant differences in antibody levels. ns, nonsignificant; *, $P < 0.05$; **, $P < 0.01$; ***, $P < 0.001$.

M0 and M1 and subsequently in several screenings at the Hospital Clínic and any time the participant had symptoms or had been in contact with a SARS-CoV-2-infected person. Blood samples were collected at M0, M1, and M6 by venipuncture and at M3 by finger prick to collect plasma and perform serological tests. Detailed characteristics of the study cohort have been previously described (3, 8, 25). None of the participants from the Barcelona cohort had been vaccinated at any of the time points tested.

**Samples and data collection.** In both study cohorts, plasma (Barcelona) and serum (Munich) samples were isolated and cryopreserved at −80°C until tested. Participants enrolled in each cohort were asked to complete a standardized electronic questionnaire in REDCap (Research Electronic Data Capture) version 8.8.2 (53) for the Barcelona cohort and in the open-source electronic case form system m4 DIS (BitCare GmbH; https://www.bitcare.de) (54) for the Munich cohort. The following information was collected from both cohorts, homogenized, and merged to be used in the study: demographics (age, sex), professional information (occupation at the hospital, daily contact with COVID-19 patients), clinical information (baseline illness, chronic medication), history of SARS-CoV-2 rRT-PCR testing, COVID-19 symptoms (cough, sore throat, rhinorrhea, fatigue, dyspnea, fever, headache, emesis, diarrhea, loss of smell, loss of taste, and rigors), and smoking habits.

**rRT-PCR.** SARS-CoV-2 detection by rRT-PCR was based on the nucleocapsid gene regions 1 (N1) and N2 following the CDC-006-00019 CDC/DDID/NCIRD/Division of Viral Diseases protocol as previously described (3, 8).

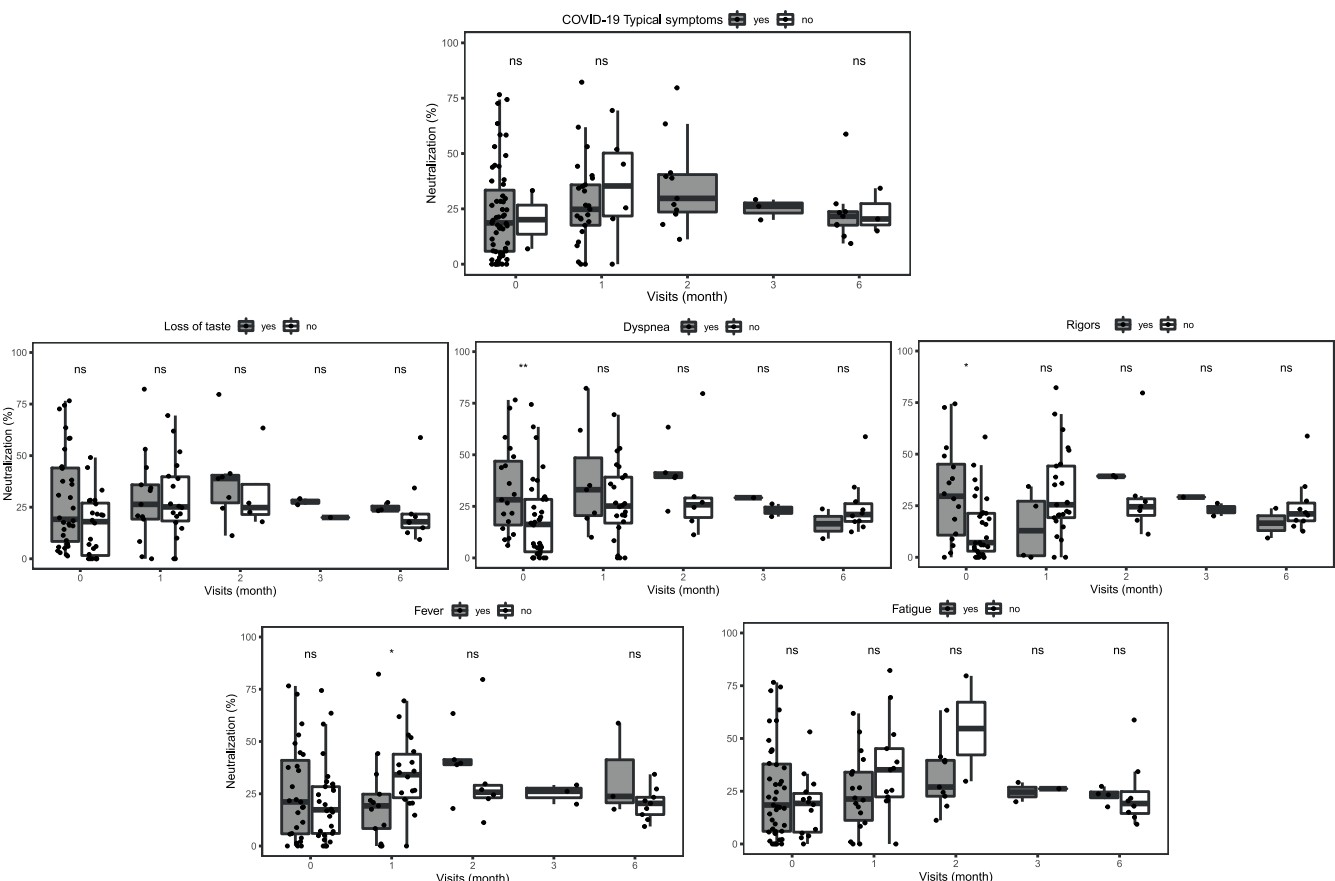

**FIG 7** Comparison of plasma neutralization activity between symptomatic and asymptomatic individuals at different time points after a positive diagnostic by rRT-PCR. (A) Comparison between symptomatic and asymptomatic ($n = 114$). (B) Comparison between presence or absence of the specific symptoms positively associated to neutralization in the multiple linear regression model (loss of taste [$n = 113$], dyspnea [$n = 113$], rigors [$n = 98$], fever [$n = 113$], fatigue [$n = 114$]). Plots show percentage of plasma neutralization activity. The center line of boxes depicts the median values, the lower and upper hinges correspond to the first and third quartiles, the distance between the first and third quartiles corresponds to the interquartile range (IQR), and whiskers extend from the hinge to the highest or lowest value within $1.5\times$ IQR of the respective hinge. Wilcoxon rank sum test was used to assess statistically significant differences in percent neutralization. ns, nonsignificant; *, $P < 0.05$; **, $P < 0.01$; ***, $P < 0.001$.

**Quantification of IgA, IgM, and IgG to SARS-CoV-2 antigens by Luminex.** The levels of IgM, IgG, and IgA were assessed in single replicates by high-throughput multiplex quantitative suspension array technology (qSAT) as previously described (25). The SARS-CoV-2 antigens included were the full-length S protein (amino acids [aa] 1 to 1213 expressed in Expi293 and His tag purified), the subunit S2 (purchased from Sino Biologicals), the RBD (donated by the Krammer Lab, Mount Sinai, New York), the full-length N protein (N-FL), and the C-terminal region (N-Cterm). Briefly, antigen-coupled beads (MagPlex polystyrene 6.5 $\mu$m COOH-microspheres, Luminex Corp., Austin, TX, USA) were added to a 384-well $\mu$Clear flat-bottom plate (Greiner Bio-One, Frickenhausen, Germany) in multiplex (2,000 microspheres per analyte per well) in a volume of 90 $\mu$L of Luminex buffer (1% bovine serum albumin [BSA], 0.05% Tween 20, 0.05% sodium azide in phosphate-buffered saline [PBS]). A hyperimmune pool at 2-fold 8 serial dilutions from 1:12.5 was used as positive control in each assay plate for quality assurance/quality control (QA/QC) purposes. Prepandemic samples were used as negative controls to estimate the cutoff of seropositivity. Ten microliters of each dilution of the positive control, negative controls, and test samples (prediluted 1:50 in 96-well round-bottom plates) were added to the 384-well plate (final test sample dilution of 1:500). To quantify IgM responses, test samples and controls were pretreated with anti-human IgG (Gullsorb) at 1:10 dilution to avoid IgG interferences.

Assay positivity cutoffs specific for each isotype and antigen pair were calculated as 10 to the mean plus 3 standard deviations (SD) of $\log_{10}$-transformed mean fluorescence intensity (MFI) of the 129 prepandemic controls. Positive serology was defined by being positive for IgG, IgA, and/or IgM to any of the antigens tested. Results were defined as indeterminate when the MFI levels for a given isotype-analyte pair were between the positivity threshold and an upper limit at 10 to the mean plus 4.5 SD of the $\log_{10}$-transformed MFIs of prepandemic samples, no other isotype-antigen combination was above the positivity cutoff, and the participant did not have any previous evidence of seropositivity or rRT-PCR positivity. Assay performance was previously established as 100% specificity and 95.78% sensitivity for seropositivity 14 days after symptom onset (55).

**Quantification of IgG to RBD by ELISA.** His-tagged RBD of SARS-CoV-2 S protein (accession number QHD43416.1) was expressed in HEK293 cells, purified using a Ni-nitrilotriacetic acid (Ni-NTA) purification system, and used as ELISA antigen. Each well of a 96-well MaxiSorp Nunc-Immuno plate (Thermo Fisher Scientific, Roskilde, Denmark) was coated with 100 ng of RBD by incubating 1 h at room temperature. Following plate washing (Wellwash Microplate Washer; Thermo Fisher Scientific, Waltham, Massachusetts) and blocking overnight at 4°C (SmartBlock; Candor Bioscience GmbH, Wangen, Germany), 1:100 diluted plasma/serum samples were incubated for 1 h at 37°C. Plates were washed and then incubated with protein A horseradish peroxidase (HRP) conjugate (Merck, Darmstadt, Germany) diluted 1:138 for 30 min at 37°C. Plates were washed again and incubated with TMB substrate (Invitrogen, Austria) during 30 min, and the reaction was stopped with 1 M phosphoric acid. Extinction at 450 nm and 650 nm was measured using a Tecan sunrise plate reader (Tecan Trading AG, Switzerland). The seropositivity cutoff from each plate was calculated by taking average optical density (OD) ($OD_{450-630}$, OD at 450 nm after background subtraction at 630 nm) of 256 negative controls (obtained from blood donors before 2020). Samples with borderline results were reanalyzed by recomLine SARS (Mikrogen) to confirm negativity and were below the final cutoff of 0.375 $OD_{450-630}$. Signal to cutoff ratio was calculated as the ratio of the $OD_{450-630}$ value from test sample to cutoff value.

**Flow cytometry-based surrogate neutralization assay.** Percentage of inhibition of RBD binding to ACE2 by plasma/serum was analyzed through a flow cytometry cell-based assay as previously described (25). This assay was validated by direct comparison of 50% inhibitory concentration ($IC_{50}$) neutralization values obtained applying a pseudovirus-based neutralization assay using HIV-based pseudovirus and ACE2 expressing 293T cells (56). Briefly, a murine stable cell line expressing the human ACE2 receptor ($1.2 \times 10^3$ 300.19-ACE2 cells per well in a 96-well plate) was incubated with RBD-mFc fusion proteins (4 $\mu$g/mL), previously exposed to plasma/serum samples at a dilution of 1:50 in PBS for 30 min at 4°C. Cells were stained with anti-mouse IgG-PE (Jackson ImmunoResearch), washed, and analyzed by Flow cytometry using standard procedures. Samples were acquired with a FACSCanto II (BD Biosciences) and analyzed with FlowJo Xv10.0.7 (Tree Star, Inc) software (57). The surrogate neutralization assay was performed in a subset of 50 and 265 samples from the Munich and Barcelona cohorts, respectively. Samples were randomly selected covering the whole range of IgG-RBD levels at the Luminex.

**Statistical analysis.** Antibody levels measured by Luminex were $log_{10}$-transformed prior to data analysis. Frequency distributions in Tables 1 and 2 are presented as means and standard deviations for continuous variables or numbers and percentages for categorical variables. Correlations of antibody levels with the plasma surrogate-neutralizing activity were assessed with the Spearman's rank correlation coefficient $\rho$ (rho).

Associations between antigen-specific IgG levels measured by ELISA or Luminex with plasma surrogate-neutralizing activity were assessed by linear regression models. Associations of COVID-19-compatible symptoms, direct contact with COVID-19 patients, comorbidities, chronic medications, and smoking habit with plasma anti-RBD IgG levels and plasma surrogate-neutralizing activity were estimated by multiple linear regression models adjusting for age and sex. Leave-one-out cross-validation (LOOCV) and root mean square error (RSME) were used to evaluate the performance of prediction models. The comparisons of antibody levels and plasma surrogate-neutralizing activity between cohorts and between symptomatic and asymptomatic subjects were represented in boxplots showing the medians and interquartile ranges (IQR), and Wilcoxon rank sum test was used to assess statistically significant differences. All analyses were performed in R (version 3.6.0) and a double-side $P$ of $<0.05$ was considered statistically significant.

**Data availability.** The raw data supporting the conclusions of this article will be made available by the authors upon request.

## SUPPLEMENTAL MATERIAL

Supplemental material is available online only.
**SUPPLEMENTAL FILE 1**, PDF file, 0.1 MB.

## ACKNOWLEDGMENTS

This work was supported by the European Institute of Innovation and Technology (EIT) Health (grant number 20877), supported by the European Institute of Innovation and Technology, a body of the European Union receiving support from the H2020 Research and Innovation Program. The study was also supported by the Institut de Salut Global de Barcelona (ISGlobal) internal funds, in-kind contributions from Hospital Clínic de Barcelona and the Fundació Privada Daniel Bravo Andreu. ISGlobal acknowledges support from the Spanish Ministry of Science and Innovation and State Research Agency through the "Centro de Excelencia Severo Ochoa 2019-2023" Program (CEX2018-000806-S) and support from the Generalitat de Catalunya through the CERCA Program. L.I. was supported by the PID2019-110810RB-I00 grant from the Spanish Ministry of Science & Innovation. R.R. had the support of the Health Department, Catalan Government (PERIS SLT017/20/000224). Development of SARS-CoV-2 reagents was partially supported by the

National Institute of Allergy and Infectious Diseases Centers of Excellence for Influenza Research and Surveillance (contract number HHSN272201400008C).

The funders had no role in study design, data collection and analysis, the decision to publish, or the preparation of the manuscript.

We thank the participation of HCW who are committed to this study and are key personnel facing the pandemic.

We are grateful for contributions to the Barcelona (SeroCov) cohort from the following participants: Diana Barrios, Laura Pujol, Rebeca Santano, Robert A. Mitchell, Chenjerai Jairoce, Cristina Castellana, Pau Cisteró, Selena Alonso, Javier Moreno, Jochen Hecht, Mikel J. Martínez, Dani Parras, Pau Serra, Pere Santamaria, Natalia Rodrigo Melero, and Jordi Chi for laboratory support; Sarah Williams, Angeline Cruz, Antía Figueroa-Romero, Neus Rosell, Patricia Sotomayor, Sara Torres, Silvia Fochs, Nuria Pey, Eugenia Chóliz, Montserrat Lamoglia, Nuria Rosell, Anna Llupià, Anna Vilella, Pilar Varela, Antoni Trilla, and Sonia Barroso for field work support; Sergi Sanz and Susana Méndez for data management support.

We are grateful to the following members of the SeCoMRI study group for their contributions to the Munich cohort: Balqees Al Darweesh, Clara Balzer, Felix Bauerdorf, Alexander Böhner, Dirk Busch, Lisena Cala, Ana Cirac, Adam Chaker, Anaïs Marie Theresa Doll, Johanna Erber, Manon Feuchtinger, Ana Galhoz, Friedemann Gebhardt, Marisa Geisberger, Markus Gerhard, Oliver Goldhardt, Katharina Gresset-Kaliebe, Natalia Graf, Florian Groß, Roman Günthner, Martin Halle, Bernhard Haller, Joachim Hellemann, Andreas Henkel, Maximilian Hinz, Dieter Hoffmann, Klaus-Peter Janssen, Robert Kaczmarczyk, Verena Kappler, Percy Knolle, Florian Kohlmayer, Susanne Kossatz, Klaus Kuhn, Zsuzsanna Kurgyis, Vincent Lallinger, Judith Lammer, Paul Lingor, Elke Lorenz, Felix Mayr, Michael M. Menden, Hrvoje Mijočević, Caroline Sandra Moesta, Ruth Neuhauser, Andrea Pagani, Anna Caroline Pilz, Clarissa Prazeres da Costa, Sarah Preis, Ulrike Protzer, Michael Quante, Hedwig Roggendorf, Jürgen Ruland, Cora Scheerer, Roland M. Schmid, Paul Schmidle, Christine Schönmann, Florian Schraml, Christoph D. Spinner, Annette Susanne Steimle-Grauer, Christian Stöß, Pavel Stupakov, Markus Thaler, Dolores Thum, Dirk Tomsitz, Wolfgang Weber, Angelika Werner, and Christof Winter.

We declare that the research was conducted in the absence of any commercial or financial relationships that could be construed as a potential conflict of interest.

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
