## [Reviewer comments · Microbiology Spectrum]

Microbiology Spectrum

RBD-based ELISA and Luminex predict anti-SARS-CoV-2 surrogate-neutralizing activity in two longitudinal cohorts of German and Spanish health care workers

Ruth Aguilar, Xue Li, Claudia Crowell, Teresa Burrell, Marta Vidal, Rocio Rubio, Alfons Jiménez, Pablo Hernández-Luis, Dieter Hoffmann, Hrvoje Mijočević, Samuel Jeske, Catharina Christa, Elvira D'Ippolito, Paul Lingor, Percy Knolle, Hedwig Roggendorf, Alina Priller, Sarah Yazici, Carlo Carolis, Alfredo Mayor, Patrik Schreiner, Holger Poppert, Henriette Beyer, Sophia Schambeck, Luis Izquierdo, Marta Tortajada, Ana Angulo, Erwin Soutschek, Pablo Engel, Alberto García-Basteiro, Dirk Busch, Gemma Moncunill, Ulrike Protzer, Carlota Dobaño, and Markus Gerhard

Corresponding Author(s): Markus Gerhard, Technische Universität München, and Carlota Dobaño, Instituto de Salud Global Barcelona

Review Timeline:

Submission Date:	August 12, 2022
Editorial Decision:	September 12, 2022
Revision Received:	November 30, 2022
Accepted:	December 4, 2022

Editor: Maria Grazia Cusi

Reviewer(s): The reviewers have opted to remain anonymous.

Transaction Report:

DOI: <https://doi.org/10.1128/spectrum.03165-22>

Dr. Markus Gerhard
Technical University of Munich and German Center for Infection Research
Munich
Germany

Re: Spectrum03165-22 (RBD-based ELISA and Luminex predict anti-SARS-CoV-2 plasma surrogate-neutralizing activity in two longitudinal cohorts of German and Spanish health care workers)

Dear Dr. Markus Gerhard:

I have received the reviews of your manuscript entitled "RBD-based ELISA and Luminex predict anti-SARS-CoV-2 plasma surrogate-neutralizing activity in two longitudinal cohorts of German and Spanish health care workers", and I regret to inform you that we will not be able to publish it in Spectrum. Your submission was read by reviewers with expertise in the area addressed in your study and it was the consensus view of these reviewers that your paper did not meet the standards necessary for publication. Copies of the reviewers' comments are attached for your consideration.

I am sorry to convey a negative decision on this occasion, but I hope that the enclosed reviews are useful. Please note, rejections from Microbiology Spectrum are final and your manuscript will not be considered by other ASM journals. We wish you well in publishing this report in another journal and hope that you will consider Spectrum in the future.

Sincerely,

Maria Grazia Cusi
Editor, Microbiology Spectrum

Reviewer comments:

Reviewer #1 (Comments for the Author):

Aguilar et al have correlated high throughput ELISA markers of IgA, IgM and IgG to different COVID antigen targets to neutralizing responses in 2 cohorts of healthcare workers. Their results are in agreement with similar published studies indicating a correlation between IgG responses and the RBD domain.

Major comments:

- 1) Line 139. As the authors indicate, there is no standardization or harmonization of immunoassays for detecting immunological response for COVID markers. In light of this, the authors should discuss the implications of the results and conclusions to neutralization correlation with the assay used in this manuscript against that more broadly when using other immunoassays commercially available.
- 2) Line 302. Only a subset of samples were used to evaluate against the gold standard neutralization assay. How were these selected to prevent bias within the data set and conclusions drawn?
- 3) Figure 2 and 3: The authors include both negative and positive A_{by} response in their correlations against neutralization %. Have they considered repeating the analysis with the non-reactive (negative) ELISA results? Are the non-reactive samples included in the analysis influencing the linear regression? Furthermore, in figure 2, there are different percentages of negative, indeterminant, and reactive samples, which can impact the linear regression and correlation. The authors should address this.
- 4) Table 4: Can the authors comment on why antibody responses to N-FL and S2 have negative correlation to % neutralization?
- 5) Line 469. The authors indicate that some of their observations are influenced by to small of a sample size. Given this, the authors should limit conclusions where this is the case.
- 6) Figures 5-7. The authors conclude that the highest correlation between neutralization and antibody response is to the RBD antigen. They then show that RBD IgG is correlated to common symptoms of COVID. However, there is largely no statistical significant link between COVID symptoms and % neutralization. The authors should address this in more detail.

Minor comments:

1) Given the dates of enrollment of healthcare workers, it is assumed that none had been vaccinated. Can the authors confirm this, and if so, add a statement as such?

2) Line 257. Did the authors confirm removal of all IgG antibodies when testing IgM response? If so, please indicate as such and data not shown.

Reviewer #2 (Comments for the Author):

The manuscript by Aguilar et al. presents potential interesting data; however, the way they are presented is confusing and makes difficult the understanding of the study.

There are several discrepancies:

In Methods Section "Sample and data collection" (line 224), the Authors stated: "In both cohorts, plasma samples were isolated and cryopreserved at -80{degree sign}C until tested", but then under "Quantification of IgG to RBD by ELISA" paragraph (line 278), they stated "1:100 diluted serum samples were incubated..."; under "Munich cohort" paragraph (line 199): "Anonymized serum samples were obtained..."; under "Barcelona cohort" (line 219): "Blood samples were collected at M0, M1 and M6 by venipuncture and at M3 by finger prick to perform serological tests". The Authors should carefully check whether the samples are all plasma or not because the analysis performed on heterogeneous blood samples (plasma, serum and whole blood obtained by the finger prick) could significantly alter the power of the results and could probably be involved in the differences of the serological profiles found between the cohorts.

Regarding statistical analysis, in the legend of figures 1, 2, 6 and 7 the Authors correctly reported data as median values but under "statistical analysis" paragraph they mentioned only means and standard deviation (that means Gaussian distribution). However, the main limitation of the study is about the stratification of samples. Under Methods Section (lines 201-203), the Authors stated "Participants with clinical symptoms compatible with COVID-19 illness, or who had a positive SARS-CoV-2 nasopharyngeal rRT-PCR result regardless of the presence of symptoms, were included in this sub-study, in total 218 (study month 0, M0)" for the Munich cohort and 578 for the Barcelona cohort. Then, in the Discussion section (lines 415-417) "Among the study participants, some were SARS-CoV-2 convalescent individuals with mild to moderate disease and some did not report SARS-CoV-2 infection". This discordance is more evident in Table 1. It reports symptomatic and asymptomatic subjects for both cohorts and RT-PCR positive or negative samples, but it is not specified if the RT-PCR positive samples are all symptomatic subjects or a mix of symptomatic/asymptomatic, and who are the negative RT-PCR subjects.

Moreover, in Table 1 what is the meaning of "Visit times"? Are they M0, M1, M2, M3, M6 time-points or are they the number of visits made by the subjects? For example: M0, M1 and M6=3 visits; M0 and M1=2 visits? The table also shows that the sum of the subjects enrolled in the different visit times is 218 for the Munich Cohort and 578 for the Barcelona cohort, therefore the total samples of the study were taken in the 6 month interval and not all collected at baseline M0, as the Authors declared under the above reported methods section (line 203). Thus, how many of the patients at M0, M1, M2, M3 and M6 are symptomatic/asymptomatics, RT-PCR positive/negative?

Furthermore, regarding the surrogate neutralization assay, the Authors stated under the "Flow cytometry-based surrogate neutralization assay" paragraph (lines 403-404), that "the assay was performed in a subset of 60 and 265 samples from the Munich and Barcelona cohorts, respectively". How did they choose the samples? Again, how many are symptomatics, asymptomatics, RT-PCR positive/negative?

Another great limitation of this study is the lack of information about the numerosity of the samples analyzed. All the figures miss the numerosity, as a consequence, it is really difficult to evaluate the importance of data.

In Figures 2A, 2B, 2C, the combined cohort is 265+60, Barcelona 265 and Munich 60?

The same for Figure 3? To note, serology data have been reported only for Luminex in Table 2 and not for ELISA IgG RBD.

Figures 5A, 5B, 6 and 7 miss again the numerosity of the groups and in figure 6 the median values should also be reported.

Figure 7 is really illogical, because in panel A the number of general asymptomatic individuals is lower than the number of asymptomatic individuals with a single specific COVID-19 symptom, reported in panel B. Moreover, the number of samples is too low for proper statistical analysis and a correct interpretation of data, leading to probable imprecise conclusions.

Finally, the good correlation between serological data and neutralization assays have already been reported and published by several studies. Therefore, as a main focus, the Authors should analyze the (homogeneous) data obtained by Luminex or ELISA compared to the surrogate neutralization assays, trying to provide a serological cutoff for the specific anti-SARS-CoV-2 antibodies level able to protect from the infection.

The manuscript by Aguilar et al. presents potential interesting data; however, the way they are presented is confusing and makes difficult the understanding of the study.

There are several discrepancies:

In Methods Section “Sample and data collection” (line 224), the Authors stated: “In both cohorts, plasma samples were isolated and cryopreserved at -80°C until tested”, but then under “Quantification of IgG to RBD by ELISA” paragraph (line 278), they stated “1:100 diluted serum samples were incubated...”; under “Munich cohort” paragraph (line 199): “Anonymized serum samples were obtained...”; under “Barcelona cohort” (line 219): “Blood samples were collected at M0, M1 and M6 by venipuncture and at M3 by finger prick to perform serological tests”. The Authors should carefully check whether the samples are all plasma or not because the analysis performed on heterogeneous blood samples (plasma, serum and whole blood obtained by the finger prick) could significantly alter the power of the results and could probably be involved in the differences of the serological profiles found between the cohorts.

Regarding statistical analysis, in the legend of figures 1, 2, 6 and 7 the Authors correctly reported data as median values but under “statistical analysis” paragraph they mentioned only means and standard deviation (that means Gaussian distribution).

However, the main limitation of the study is about the stratification of samples. Under Methods Section (lines 201-203), the Authors stated “Participants with clinical symptoms compatible with COVID-19 illness, or who had a positive SARS-CoV-2 nasopharyngeal rRT-PCR result regardless of the presence of symptoms, were included in this sub-study, in total 218 (study month 0, M0)” for the Munich cohort and 578 for the Barcelona cohort. Then, in the Discussion section (lines 415-417) “Among the study participants, some were SARS-CoV-2 convalescent individuals with mild to moderate disease and some did not report SARS-CoV-2 infection”. This discordance is more evident in Table 1. It reports symptomatic and asymptomatic subjects for both cohorts and RT-PCR positive or negative samples, but it is not specified if the RT-PCR positive samples are all symptomatic subjects or a mix of symptomatic/asymptomatic, and who are the negative RT-PCR subjects.

Moreover, in Table 1 what is the meaning of “Visit times”? Are they M0, M1, M2, M3, M6 time-points or are they the number of visits made by the subjects? For example: M0, M1 and M6=3 visits; M0 and M1=2 visits? The table also shows that the sum of the subjects enrolled in the different visit times is 218 for the Munich Cohort and 578 for the Barcelona cohort, therefore the total samples of the study were taken in the 6 month interval and not all collected at baseline M0, as the Authors declared under the above reported methods section (line 203). Thus, how many of the patients at M0, M1, M2, M3 and M6 are symptomatic/asymptomatic, RT-PCR positive/negative?

Furthermore, regarding the surrogate neutralization assay, the Authors stated under the “Flow cytometry-based surrogate neutralization assay” paragraph (lines 403-404), that “the assay was performed in a subset of 60 and 265 samples from the Munich and Barcelona cohorts, respectively”. How did they choose the samples? Again, how many are symptomatics, asymptomatics, RT-PCR positive/negative?

Another great limitation of this study is the lack of information about the numerosity of the samples analyzed. All the figures miss the numerosity, as a consequence, it is really difficult to evaluate the importance of data.

In Figures 2A, 2B, 2C, the combined cohort is 265+60, Barcelona 265 and Munich 60?

The same for Figure 3? To note, serology data have been reported only for Luminex in Table 2 and not for ELISA IgG RBD.

Figures 5A, 5B, 6 and 7 miss again the numerosity of the groups and in figure 6 the median values should also be reported.

Figure 7 is really illogical, because in panel A the number of general asymptomatic individuals is lower than the number of asymptomatic individuals with a single specific COVID-19 symptom, reported in panel B. Moreover, the number of samples is too low for proper statistical analysis and a correct interpretation of data, leading to probable imprecise conclusions.

Finally, the good correlation between serological data and neutralization assays have already been reported and published by several studies. Therefore, as a main focus, the Authors should analyze the (homogeneous) data obtained by Luminex or ELISA compared to the surrogate neutralization assays, trying to provide a serological cutoff for the specific anti-SARS-CoV-2 antibodies level able to protect from the infection.

Spectrum03165-22_Response to reviewers' comments

Reviewer #1 (Comments for the Author):

Aguilar et al have correlated high throughput ELISA markers of IgA, IgM and IgG to different COVID antigen targets to neutralizing responses in 2 cohorts of healthcare workers. Their results are in agreement with similar published studies indicating a correlation between IgG responses and the RBD domain.

Major comments:

1) Line 139. As the authors indicate, there is no standardization or harmonization of immunoassays for detecting immunological response for COVID markers. In light of this, the authors should discuss the implications of the results and conclusions to neutralization correlation with the assay used in this manuscript against that more broadly when using other immunoassays commercially available.

Answer: Thank you for your comments. We have expanded the text in the discussion to acknowledge this point (pages 14 and 15, lines 508-517): "Although there is no common reference assay for SARS-CoV-2 serology worldwide, and correlations between in-house and commercial ELISA- or Luminex-based assays and neutralization activity may slightly differ between laboratories depending on reagents and procedures used, it is remarkable that results are consistent for RBD IgG. To improve comparability among surrogate virus neutralization assays from different origins, it is critical to establish international units per milliliter (IU/mL) for SARS-CoV-2 serum/plasma neutralizing activity. Since the performance of our assays was very robust, we are however confident that they can accurately be used as surrogates of neutralizing activity despite the slight differences that there might be with other similar manufactured assays."

2) Line 302. Only a subset of samples was used to evaluate against the gold standard neutralization assay. How were these selected to prevent bias within the data set and conclusions drawn?

Answer: Samples were randomly selected covering the whole range of IgG-RBD levels measured by Luminex, so they were representative of different levels of response. This has now been added to the Methods section (page 9 lines 311-312).

3) Figure 2 and 3: The authors include both negative and positive A_{by} response in their correlations against neutralization %. Have they considered repeating the analysis with the non-reactive (negative) ELISA results? Are the non-reactive samples included in the analysis influencing the linear regression? Furthermore, in figure 2, there are different percentages of negative, indeterminate, and reactive samples, which can impact the linear regression and correlation. The authors should address this.

Answer: The approach to include a broad range of IgG reactivities in the assessment of the correlation with nAb data gives a more reliable performance of the assays and correlations. However, as suggested by the reviewer, we have checked the correlations including only positive and indeterminate measurements (the indeterminate are positive results between the positivity cut off [mean + 3SD] and mean + 4.5SD above). These new correlations are now in Supplementary Tables 1 and 2, and results show that almost all positive correlations in Figure 2 and all in Figure 3 are maintained. The main differences are for IgM-S vs. % neutralization with no correlation in Barcelona samples when discarding the negative ones, but showing a moderate correlation in Munich samples that was not observed when including the

negative samples. IgA-S2 also loses the correlation with % neutralization in Munich samples when excluding the negative ones, however the sample size is small. Other correlations affected were those of N antigens, probably related to the high proportion of negative samples against this antigens and the presence of cross-reactive anti-huCov antibodies in the positive ones with low neutralization capacity.

4) Table 4: Can the authors comment on why antibody responses to N-FL and S2 have negative correlation to % neutralization?

Answer: The nAb assay used in the study is RBD-based, and N and S2 do not contain RBD, thus higher antibody levels against N and S2 are not expected to increase the % of neutralization. However we do not have an explanation for the negative correlation of these antibodies against the % neutralization. They could be interfering with the anti-RBD response, or indirectly associated to a lower neutralizing activity. This has now been added in page 13 lines 460-466.

5) Line 469. The authors indicate that some of their observations are influenced by to small of a sample size. Given this, the authors should limit conclusions where this is the case.

Answer: The small sample size only affects the analyses with the subgroup of participants with rRT-PCR positive results and neutralization data, but it does not affect the analyses with all the study participants with neutralization data, thus our main conclusion that serology of IgG to RBD is a good surrogate of % neutralization is not affected. This has now been added to the discussion (page 14 lines 501-506)

6) Figures 5-7. The authors conclude that the highest correlation between neutralization and antibody response is to the RBD antigen. They then show that RBD IgG is correlated to common symptoms of COVID. However, there is largely no statistical significant link between COVID symptoms and % neutralization. The authors should address this in more detail.

Answer: Analyses in figure 7 are performed only with subjects rRT-PCR positive and data are stratified by symptoms and time point. This reduces the sample size and thus the statistical power of the comparisons. However, in spite of this, at baseline, where the sample size is higher, dyspnea and rigors are associated to higher % of neutralization, and trends are observed for other symptoms which suggest that it is a matter of small samples size, and that COVID-19 symptoms are probably associated to % neutralization.

Minor comments:

1) Given the dates of enrollment of healthcare workers, it is assumed that none had been vaccinated. Can the authors confirm this, and if so, add a statement as such?

Answer: Correct, none of the healthcare workers had been vaccinated. A statement has been added in page 7 lines 225-226.

2) Line 257. Did the authors confirm removal of all IgG antibodies when testing IgM response? If so, please indicate as such and data not shown.

Answer: It is not possible to remove all IgG, or at least very difficult. The Gullisorb treatment used in the study improves IgM measurements by sequestering most of the IgG. This has been tested in previous experiments but we do not re-test it again in each new study. We have numerous publications, using this protocol.

Some examples:

- Eleven-month longitudinal study of antibodies in SARS-CoV-2 exposed and naïve primary health care workers upon COVID-19 vaccination. Dobaño, Ramírez-Morros, Alonso, et al., 2022, Immunology. doi: 10.1111/imm.13551
- Highly Sensitive and Specific Multiplex Antibody Assays To Quantify Immunoglobulins M, A, and G against SARS-CoV-2 Antigens. Dobaño, Vidal, Santano et al., 2021. J Clin Microbiol. doi: 10.1128/JCM.01731-20
- Sustained seropositivity up to 20.5 months after COVID-19. Dobaño, Ramírez-Morros, Alonso et al., 2022 BMC Medicine doi: 10.1186/s12916-022-02570-3

Reviewer #2 (Comments for the Author):

The manuscript by Aguilar et al. presents potential interesting data; however, the way they are presented is confusing and makes difficult the understanding of the study.

There are several discrepancies:

In Methods Section "Sample and data collection" (line 224), the Authors stated: "In both cohorts, plasma samples were isolated and cryopreserved at -80°C until tested", but then under "Quantification of IgG to RBD by ELISA" paragraph (line 278), they stated "1:100 diluted serum samples were incubated..."; under "Munich cohort" paragraph (line 199): "Anonymized serum samples were obtained..."; under "Barcelona cohort" (line 219): "Blood samples were collected at M0, M1 and M6 by venipuncture and at M3 by finger prick to perform serological tests". The Authors should carefully check whether the samples are all plasma or not because the analysis performed on heterogeneous blood samples (plasma, serum and whole blood obtained by the finger prick) could significantly alter the power of the results and could probably be involved in the differences of the serological profiles found between the cohorts.

Answer: Barcelona samples were plasma and Munich samples were serum. However, for antibody determinations the impact of the anticoagulant is minimal. We have now indicated in the Methods section whether samples were plasma or serum depending on the cohort (page 7 line 229).

Some references showing that serum and plasma antibody titers correlate:

- Correlation between Serum and Plasma Antibody Titers to Mycobacterial Antigens. Siev, Yu, Prados-Rosales, 2011. Clin Vaccine Immunol. doi: 10.1128/CVI.00325-10
- Comparison of the Use of Serum and Plasma as Matrix Specimens in a Widely Used Noncommercial Dengue IgG ELISA. Deza-Cruz, Mill, Rushton and Kelly 2019, Am J Trop Med Hyg. doi: 10.4269/ajtmh.19-0112
- Stability of SARS-CoV-2 IgG in multiple laboratory conditions and blood sample types. NKanji, Bailey, Fenton et al. 2021. Journal of Clinical Virology. doi: 10.1016/j.jcv.2021.104933

Regarding statistical analysis, in the legend of figures 1, 2, 6 and 7 the Authors correctly reported data as median values but under "statistical analysis" paragraph they mentioned only means and standard deviation (that means Gaussian distribution).

Answer: Means and SD are reported for the frequency distributions of continuous variables in Tables 1 and 2, while antibody levels and % nAb are represented by boxplots with the medians and the IQR as explained in the legend figures. We have now added a paragraph in the methods explaining the variables and data represented in boxplots (page 10 lines 328-332).

However, the main limitation of the study is about the stratification of samples. Under Methods Section (lines 201-203), the Authors stated "Participants with clinical symptoms compatible with COVID-19 illness, or who had a positive SARS-CoV-2 nasopharyngeal rRT-PCR result regardless of the presence of symptoms, were included in this sub-study, in total 218 (study month 0, M0)" for the Munich cohort and

578 for the Barcelona cohort. Then, in the Discussion section (lines 415-417) "Among the study participants, some were SARS-CoV-2 convalescent individuals with mild to moderate disease and some did not report SARS-CoV-2 infection".

Answer: Recruitment was very different between cohorts as indicated in the Methods. In Munich, participants recruited were rRT-PCR positive or symptomatic, while in Barcelona cohort were HCW randomly selected, thus most of them were non-infected. Even though stratification was a limitation, it did not affect conclusions of the study.

This discordance is more evident in Table 1. It reports symptomatic and asymptomatic subjects for both cohorts and RT-PCR positive or negative samples, but it is not specified if the RT-PCR positive samples are all symptomatic subjects or a mix of symptomatic/asymptomatic, and who are the negative RT-PCR subjects.

Answer: Stratifications by symptoms yes/no and rRT-PCR pos/neg were independent, thus we may have rRT-PCR positive samples that are from asymptomatic participants, and all other possible combinations. As for who were the rRT-PCR negative subjects, they were mostly from the Barcelona cohort but some were also from the Munich cohort. The purpose of this table was to give an overview of the type of participants we have regarding symptoms and rRT-PCR data, all stratified by cohort because of the different recruitments in each of them. Moreover, we think that this heterogeneity of the study population regarding symptoms and rRT-PCR results is not a limitation but a strength of the study for its representability of total population.

Moreover, in Table 1 what is the meaning of "Visit times"? Are they M0, M1, M2, M3, M6 time-points or are they the number of visits made by the subjects? The timepoints For example: M0, M1 and M6=3 visits; M0 and M1=2 visits?

Answer: Visit times in Table 1 refer to the number of samples from different time-points we have for each participant. Thus, in the Barcelona cohort most of the participants had 3 samples; while in Munich cohort most of the participants had 1 or 2 samples. This has now been added to the footnote of the table.

The table also shows that the sum of the subjects enrolled in the different visit times is 218 for the Munich Cohort and 578 for the Barcelona cohort, therefore the total samples of the study were taken in the 6 month interval and not all collected at baseline M0, as the Authors declared under the above reported methods section (line 203).

Answer: The total participants of the study were recruited all at baseline (M0), but the total of samples was collected in a 6 month interval. This has now been clarified in line 206.

Thus, how many of the patients at M0, M1, M2, M3 and M6 are symptomatic/asymptomatics, RT-PCR positive/negative?

Answer: We have not done this stratification by time-point and we do not think it is necessary for the purpose of the study. As indicated above, the purpose of Table 1 is to give an overview of the number of samples we have per participant, and the type of participants regarding symptoms and rRT-PCR data in each cohort.

Furthermore, regarding the surrogate neutralization assay, the Authors stated under the "Flow cytometry-based surrogate neutralization assay" paragraph (lines 403-404), that "the assay was performed in a subset of 60 and 265 samples from the Munich and

Barcelona cohorts, respectively". How did they choose the samples? Again, how many are symptomatics, asymptomatics, RT-PCR positive/negative?

Answer: Samples were randomly selected covering the whole range of IgG-RBD levels at the Luminex so they were representative of different levels of response. This has now been added to the Methods section (page 9 lines 311-312). We have not assessed the % of symptomatic and rRT-PCR positive among the selected samples because we do not think it is relevant for the purpose of the study.

Another great limitation of this study is the lack of information about the numerosity of the samples analyzed. All the figures miss the numerosity, as a consequence, it is really difficult to evaluate the importance of data.

Answer: We have now added the number of samples analyzed in each figure in the figure legends.

In Figures 2A, 2B, 2C, the combined cohort is 265+60, Barcelona 265 and Munich 60?

Answer: The samples size of the combined cohort is 265+50, 265 samples from Barcelona and 50 from Munich. This information has now been added to the figure legend.

The same for Figure 3?

Answer: In Figure 3A we have data from 710 samples, 266 being from Barcelona and 444 from Munich cohort. In figure 3B we have data from 315 samples, 265 being from Barcelona and 50 from Munich cohort. This information has now been added to the figure legend.

To note, serology data have been reported only for Luminex in Table 2 and not for ELISA IgG RBD.

Answer: This is because the purpose of table 2 is to report the proportion of SARS-CoV-2 seropositive participants in each study cohort at each of the follow up visits, and the Luminex assay is more sensitive than ELISA to detect seropositivity because it measures three isotypes (IgG, IgA and IgM) against 5 antigens (N-FL, N-Cterm, S, RBD and S2), while the ELISA only measures IgG to RBD.

Figures 5A, 5B, 6 and 7 miss again the numerosity of the groups and in figure 6 the median values should also be reported.

Answer: Samples size in each figure has now been added to the figures legends. In figure 6 the center line of boxes depicts the median values.

Figure 7 is really illogical, because in panel A the number of general asymptomatic individuals is lower than the number of asymptomatic individuals with a single specific COVID-19 symptom, reported in panel B.

Answer: This is because in panel A we have included those individuals which were 100% asymptomatic (this is: with none of the symptoms compatible with COVID-19); while in panel B the "asymptomatic" are those individuals without the specific symptom reported (but they could have other symptoms). It is explained in the figure legend; **B) Comparison between presence or absence of the specific symptoms positively associated to neutralization in the multiple linear regression model.**

Moreover, the number of samples is too low for proper statistical analysis and a correct interpretation of data, leading to probable imprecise conclusions.

Answer: We already mentioned that the low sample size is a limitation for some analyses in the results and discussion. However, in spite of the small sample size we do detect some significant differences for dyspnea and rigors showing association with higher % of neutralization. This confirms that the number of samples was not too low, and that is why we included the figure in the manuscript. However, we have added a sentence in the discussion indicating that the low sample size may have caused missing other associations (page 14 lines 501-503).

Finally, the good correlation between serological data and neutralization assays have already been reported and published by several studies. Therefore, as a main focus, the Authors should analyze the (homogeneous) data obtained by Luminex or ELISA compared to the surrogate neutralization assays, trying to provide a serological cutoff for the specific anti-SARS-CoV-2 antibodies level able to protect from the infection.

Answer: This is not possible because this study was not designed to assess a serological cutoff associated to protection, which many groups worldwide are trying to obtain but is very challenging. This was not the scope of the study. *Microbial Spectrum* policy is to publish manuscripts that meet the criteria of scientific and methodological rigor regardless of potential impact.

December 4, 2022

Dr. Markus Gerhard
Technische Universitat Munchen
Munich
Germany

Re: Spectrum03165-22R1-A (RBD-based ELISA and Luminex predict anti-SARS-CoV-2 surrogate-neutralizing activity in two longitudinal cohorts of German and Spanish health care workers)

Dear Dr. Markus Gerhard:

Your manuscript has been accepted, and I am forwarding it to the ASM Journals Department for publication. You will be notified when your proofs are ready to be viewed.

Sincerely,

Maria Grazia Cusi
Editor, Microbiology Spectrum

Journals Department
Supplemental file 1: Accept